# Associations Between Feeding Behaviors, Residual Feed Intake, and Residual Average Daily Gain in Performance Tested Yearling Bulls and Heifers Fed a High-Forage Diet

**DOI:** 10.3390/ani15243574

**Published:** 2025-12-12

**Authors:** Tylor J. Yost, Nathan E. Blake, Ida Holásková, Domingo J. Mata-Padrino, John K. Yost, Jarred W. Yates, Matthew E. Wilson

**Affiliations:** 1School of Agriculture and Food Sciences, West Virginia University, Morgantown, WV 26506, USA; 2West Virginia Agricultural and Forestry Experiment Station, Office of Statistics and Data Analytics, West Virginia University, Morgantown, WV 26506, USA; 3OSU Extension Wayne County, The Ohio State University, Wooster, OH 44691, USA; yost.77@osu.edu

**Keywords:** beef cattle, feeding behavior, feed efficiency

## Abstract

It is critical that cattle producers understand the relationship between animal behavior and their growth and performance. Feedlot cattle, which are considered feed efficient or inefficient, have different characteristic eating behaviors when consuming a high-energy diet. We used 290 yearling Angus cattle to measure individual feed efficiency measures and feeding behaviors when fed a high-forage diet. Associations were found between behaviors and efficiency performance metrics confirming that efficient cattle consume feed as defined meals, while inefficient cattle have a variable eating pattern whether they are fed a high-energy or high-forage diet. Integrating behavioral differences into management practices offers a pathway to improving the sustainability of beef cattle production.

## 1. Introduction

Understanding the day-to-day variation in feed intake among individual cattle is essential for improving both production efficiency and sustainability. Feed intake can fluctuate due to management factors such as bunk design and feeding schedule [1], physiological processes related to digestion [2], differences in pen-mate uniformity [3], and animal health status [4]. Beyond these environmental and physiological influences, genetic variation in feed efficiency and growth offers producers the opportunity to select animals that not only perform better but also contribute to more sustainable production systems and improved economic returns [5,6,7].

Feeding behavior has emerged as a key component in understanding individual differences in feed efficiency. Previous research has shown that behavioral patterns at the feed bunk—such as frequency, duration, and rate of intake—can influence an animal’s efficiency [8]. However, most of these studies have been conducted using cattle fed high-concentrate diets [9,10]. Comparatively little is known about how feeding behaviors relate to efficiency measures in animals consuming high-forage diets [11], which are more representative of grazing or backgrounding systems.

Variation in feeding behavior has also been linked to differences in residual feed intake (RFI), an important measure of feed efficiency that identifies animals consuming less feed than expected for their growth and body size [12]. A better understanding of how feeding behaviors relate to both RFI and residual average daily gain (RADG) could provide valuable insights for identifying cattle that are more efficient in utilizing feed resources.

Therefore, the objective of this study was to evaluate the relationships among feeding behaviors, RFI, and RADG in performance-tested cattle fed a high-forage diet. By exploring these relationships, this study aims to enhance the understanding of behavioral and physiological drivers of efficiency, ultimately supporting the selection of more sustainable and economically efficient cattle.

## 2. Materials and Methods

### 2.1. Animal Care and Use Committee Statement

The animal work described herein was approved by the West Virginia University Animal Care and Use Committee as protocol 1608003693. The West Virginia University Animal Care and Use Committee uses the Guide for the Care and Use of Agricultural Animals for its Agricultural Animal Program. The West Virginia University Agricultural Animal Program is AAALAC accredited (#1957).

### 2.2. Animal Breeds and Sourcing

Animals used for this study were housed at West Virginia University’s Reymann Memorial Farm (Wardensville, WV, USA). The animals consisted of two groups, one of yearling purebred Angus bulls (*n* = 232) and one of yearling purebred Angus heifers (*n* = 58) being evaluated for performance as part of a separate research project. The cattle used in this study originated from 22 farms across West Virginia, Virginia, and New York. Bulls were tested during the summer months of June 2020 to August 2020, July 2021 to September 2021, July 2022 to September 2022, and the winter months November 2022 to February 2023. Heifers were tested during the summer months June 2020 to August 2020, and July 2022 to September 2022. Upon arrival, all cattle were treated with Safe-Guard^®^ (Merck Animal Health, Madison, NJ, USA) to deworm, Draxxin^®^ (Zoetis, Parsippany-Troy Hills, NJ, USA) for the prevention of bovine respiratory disease, an intranasal dose of pasteurella vaccine (Bovilis Once PMH, Merck Animal Health, Omaha, NE, USA), weighed, and ear tagged with a radio frequency identification (RFID). Upon arrival, the mean ± SEM weight of the yearling Angus bulls was 350.3 ± 3.6 kg, and of the heifers was 287.5 ± 5.0 kg. The mean age of the bulls and heifers upon arrival was 288.1 ± 1.73 days and 300.28 ± 4.2 days, respectively.

### 2.3. Collection of Intake Data

Individual feed intake was determined utilizing GrowSafe Feed Intake Nodes (GrowSafe 8000; Vytelle, Inc., Lenexa, KS, USA). Feed Intake Nodes collect feed consumption data by continuous feed bunk weighing via load cells. To access the feed bunk, an animal must pass its head through adjustable bars that limit access to one animal at a time while an antenna in the lip of the bunk reads the animal’s RFID tag [13], which is recorded as a feeding event in the data frame. The raw data of individual events from the GrowSafe units were compiled and pre-processed by Vytelle, Inc. All animals were housed in a drylot facility containing five pens measuring 29 m × 51 m with 6 feed intake nodes per pen. Bulls were grouped based on consignor group, size, and breed [13].

The performance evaluation periods for bulls and heifers followed a 14-day diet acclimation period. Bulls and heifers were fed a total mixed ration ad libitum. An average of 76 bulls were on test for an average of 71 days during the summer months of June 2020 to August 2020, June 2021 to September 2021, July 2021 to September 2021, July 2022 to September 2022, and the winter months November 2022 to February 2023. The bull ration was formulated for a target average daily gain (ADG) of 1.47 kg/day using standards for growing bulls [14] and consisted of 65% corn silage, 22.5% supplement (cotton seed hulls, dried distillers grains, soybean hulls, soybean meal, peanut hulls, and wheat middlings), 12.5% mixed grass hay (2.5–5 cm particle length), and a commercial vitamin and mineral mix containing selenium. Rumensin^®^ and Tylan^®^ (Elanco Animal Health, Indianapolis, IN, USA) were added at the labeled dosage rate. The ration contained 13% crude protein (CP) and 68% total digestible nutrients (TDN) with an average dry matter (DM) of 52%. Calculated net energy for maintenance (NE_m_) and net energy for gain (NE_g_) were 6.53 and 4.06 MJ/kg, respectively. The heifers were on test for an average of 56 days during the summer months June 2020 to August 2020, and July 2022 to September 2022. The heifer ration consisted of 80% corn silage, 14.5% supplement, and 5.5% mixed grass hay (2.5–5 cm particle length), including a commercial vitamin and mineral mix containing selenium. Rumensin^®^ and Tylan^®^ were added at the labeled dosage rate. The ration contained 11% CP and 65.3% TDN on a 44% dry matter (DM) basis. Calculated NE_m_ and NE_g_ were 6.99 and 4.51 MJ/kg, respectively. Ration samples were collected into 1 L bags from the feed truck after mixing and prior to dispensing into the feed bunks as per Cumberland Valley Analytical sample submission guidelines. Collected samples were refrigerated and shipped to Cumberland Valley Analytical Services (Waynesboro, PA, USA). Nutrient content of the ration was analyzed via Near Infrared Reflectance spectroscopy. Supplementary white salt was provided ad libitum in each pen during each evaluation period. Bulls were weighed using a conventional livestock scale every two weeks while on the test.

### 2.4. Analysis

The workflow used to analyze relationships between performance and feeding behaviors was JMP (JMP®, Version 19. JMP Statistical Discovery LLC, Cary, NC, USA). The datasets for bulls and heifers were analyzed separately. The two datasets consisted of two performance measurements; residual feed intake (RFI) and residual average daily gain (RADG), two intake measurements; average daily feed intake, and average feed consumed for each visit, and three different feeding behaviors; headdown, feeding event duration, number of visits for the entire testing period. From these, additional feeding behavior variables were calculated, such as average feed consumed per visits and per day and the average amount of visits per day. The six feeding behaviors were then tested for their correlation to the two growth measurements of RFI and RADG for both yearling bulls and yearling heifers, using simple unadjusted correlations.

### 2.5. Measurements Evaluated

Eight criteria were evaluated, two performance metrics, and six behavioral measurements. The six behavioral measurements were evaluated in this study: feeding duration, headdown, amount of feed consumed per visit, total amount consumed per day, total number of visits, and visits per day. Duration refers to the total amount of time, in seconds, that an individual animal spends at the feed bunk while its RFID tag is being read. Headdown represents the amount of time the animal is at the bunk with a short read range, which occurs because the RFID reader is located in the lip of the bunk, and is an estimation of the time they are actively consuming feed. Consumed per visit indicates the total amount of feed an animal eats per visit while consumed per day indicates how much animal eats on a daily basis during a test period. The total number of visits and visits per day measures how often an animal visits the bunk throughout the study and on a daily basis. In addition to these behavioral traits, the two performance measurements were also evaluated: residual feed intake (RFI) and residual average daily gain (RADG). RFI is defined as the difference between an individual’s actual feed intake and its expected intake based on growth rate, body composition, and weight during the testing period. RADG represents the difference between an individual’s actual and predicted average daily gain. For this study, RADG expected progeny differences (EPDs) from the American Angus Association were used.

## 3. Results

Performance metrics for the 290 animals evaluated are summarized in Table 1 and Table 2. Bulls averaged a test start weight of 350.35 ± 3.58 kg and an end test weight of 482.37 ± 3.75 kg, while the heifers averaged 278.48 ± 5.01 kg on test and 345.64 ± 4.89 kg. Bulls consumed 239.80 ± 5.17 g of feed per visit while visiting the feed bunk on average of 86.92 ± 1.44 visits per day with a duration length of 102.96 ± 2.58 s per visit. Heifers consumed 195.71 ± 10.61 g of feed per visit while visiting the feed bunk on average of 99.98 ± 5.02 visits per day with a duration length of 118.09 ± 10.58 s per visit. Bulls visited the feed bunk 6660.91 ± 122.29 times, on average, for the 76-day test period while the heifers visited the feed bunk an average of 5773.79 ± 329.46 during a 56-day period. Table 1 and Table 2 depict the performance metrics of bulls and heifers. Means were calculated by compiling data from the Vytelle input sheets and data collected by the GrowSafe feed bunks.

Figure 1 depicts the correlation matrix between feeding events and the performance measurements that were taken during the testing periods for yearling bulls. For yearling bulls, positive correlations were observed between headdown and the duration of time spent at the bunk (r = 0.85, *p* < 0.0001), individual RFI and the number of visits (r = 0.34, *p* < 0.0001), visits/day (r = 0.36, *p* < 0.0001), and the amount of feed consumed per day (r = 0.43, *p* < 0.0001). There were negative correlations between the average amount of feed consumed and the number of visits (r = −0.74, *p* < 0.0001), and between individual RADG and individual RFI (r = −0.40, *p* < 0.0001).

Figure 2 depicts the correlation matrix between feeding events and the performance measurements that were taken during a testing period for yearling Angus heifers. Positive correlations were observed between RFI and amount of feed consumed per day (r = 0.55, *p* < 0.0001), headdown and duration (r = 0.95, *p* < 0.0001). Negative correlations were between the number of visits per day and the amount consumed per day (r = −0.88, *p* < 0.0001), and between individual RFI and individual RADG (r = −0.53, *p* < 0.0001).

## 4. Discussion

Historically, individual animal feed intake was evaluated using animals housed in isolation, such as in pens or metabolism crates. Although these methods accurately measured individual dry matter intake (DMI), they did not allow researchers to observe or capture feeding behaviors that occur naturally within group settings. The development of the Calan gate system addressed this limitation by enabling the measurement of individual feed intake in group-housed animals, allowing for normal social feeding behaviors to occur [15]. Through this system, researchers and producers were able to associate feed consumption with phenotypic growth traits and make genetic selections based on superior individual performance. However, while effective for intake measurement, the Calan gate does not record specific feeding behavior traits—such as visit frequency or duration—associated with individual feeding events. This limitation restricts its ability to provide deeper insights into behavioral drivers of efficiency.

The integration of RFID-based intake monitoring systems and bunk load cells has made it possible to evaluate both passive DMI and individual feeding behaviors in more natural production environments. This technological advancement allows researchers to examine behavioral patterns in relation to feed efficiency, providing producers with more precise selection tools. The concept of feed efficiency, described by Koch et al. (1963) [6] as “gain adjusted for feed consumption”, remains critical for genetic improvement in cattle. To optimize efficiency, the industry must identify animals that can achieve high rates of gain while converting feed—particularly lower-quality forage—into high-quality protein more effectively.

Numerous studies have demonstrated that beef cattle classified as feed efficient (low-RFI) exhibit different feeding behaviors compared to feed-inefficient (high-RFI) animals when consuming high-concentrate diets [9,10,12,16]. This study expands on that knowledge by evaluating whether similar behavioral patterns exist when cattle are fed a high-forage diet.

Our results indicate that animals with fewer bunk visits but larger intakes per visit tended to be more feed efficient than those that visited the bunk more frequently but consumed less each time. These findings align with previous reports [8,9,10,16,17] and support the idea that feeding behavior can serve as a useful predictor of feed efficiency. In particular, animals that spent less total time at the bunk, yet consumed their feed more effectively during each visit, demonstrated greater efficiency. This pattern suggests that efficient animals optimize their feeding opportunities, minimizing unnecessary time at the bunk while maximizing intake per visit. These results are consistent with prior research showing that behavioral feeding traits are linked to variation in efficiency [17,18].

Bulls exhibited shorter feed consumption periods and fewer bunk visits compared with heifers, indicating clear behavioral distinctions between the two groups. These differences may be linked to underlying physiological factors, as shown in Table 1 and Table 2, where bulls consistently outweighed heifers and demonstrated higher overall feed intake. The greater body mass and growth potential of bulls likely enable them to consume larger quantities of feed in shorter periods, reducing the need for frequent bunk visits. In contrast, heifers in this study were fed a formulated ration that allowed for a lower ADG than that of the targeted gain of 1.47 kg formulated for the bulls. Collectively, these observations suggest that sex-related physiological traits play a role in shaping feeding behavior and intake dynamics.

Residual average daily gain (RADG) was also evaluated as a measure of feed efficiency. RADG represents the expected difference in growth performance among animals consuming a similar amount of feed [19]. In this study, animals with higher RADG values tended to be more efficient, similar to those with lower residual feed intake (RFI) values. However, unlike RFI, RADG did not show clear associations with specific feeding behaviors. This distinction suggests that while both RADG and RFI are useful indicators of efficiency, RFI may better capture behavioral differences that influence how efficiently animals consume and utilize their feed.

Taken together, these results highlight that feeding behavior metrics, particularly visit frequency and intake per visit, are promising indicators of feed efficiency in beef cattle. Integrating these behavioral measurements into performance testing could provide a practical and cost-effective selection tool, reducing the need for expensive intake systems. Furthermore, using behavioral traits alongside genetic predictors such as RADG may allow producers to more accurately identify efficient animals. Over time, this could improve the genetic selection for feed efficiency, reduce input costs, and enhance the sustainability of beef production systems.

## 5. Conclusions

This study provides evidence that feeding behavior traits are meaningfully associated with feed efficiency in yearling Angus cattle maintained on high-forage diets. Animals identified as more feed efficient exhibited fewer bunk visits yet consumed larger quantities per visit, suggesting that feeding consistency and reduced competitive behavior may contribute to improved nutrient utilization. The relationship observed between RFI and feeding activity patterns indicate that behavior-based metrics can serve as practical indicators of efficiency, even in forage-based production systems. These findings highlight the potential for incorporating automated feeding behavior data into selection and management programs aimed at enhancing feed efficiency and reducing input costs. Further evaluation of behavioral trends across varying environmental conditions, forage qualities, and genetic backgrounds will help refine these associations and strengthen their application in sustainable beef production.

## Figures and Tables

**Figure 1 animals-15-03574-f001:**
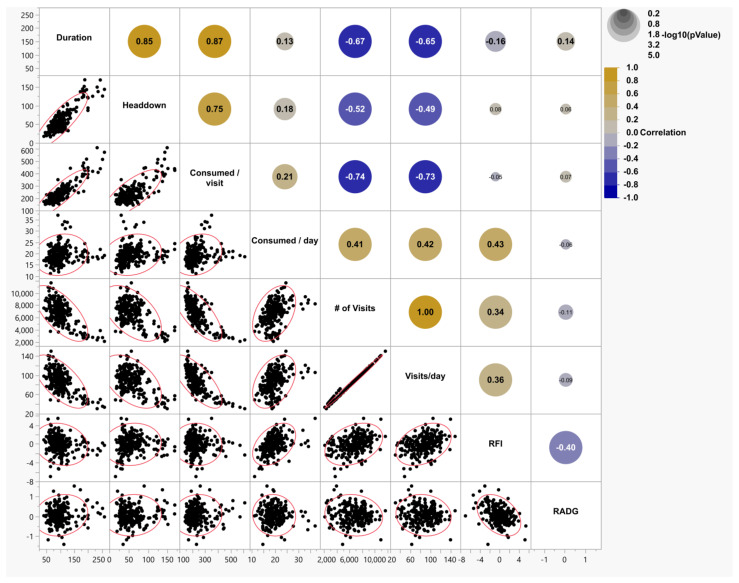
A correlation matrix representing the associations among the behavioral and efficiency variables of yearling Angus bulls (*N* = 232). In the upper right portion, the large bold values are significant simple (unadjusted) correlation coefficients at alpha = 0.05. The color gradient of the circles represents the direction of the correlations, and the size of the circles is proportional [−log 10 (*p*-value)] to the specific *p*-value. The smaller the *p*-value, the larger the circle. The lower left portion has data pairs with 95% bivariate normal density ellipses.

**Figure 2 animals-15-03574-f002:**
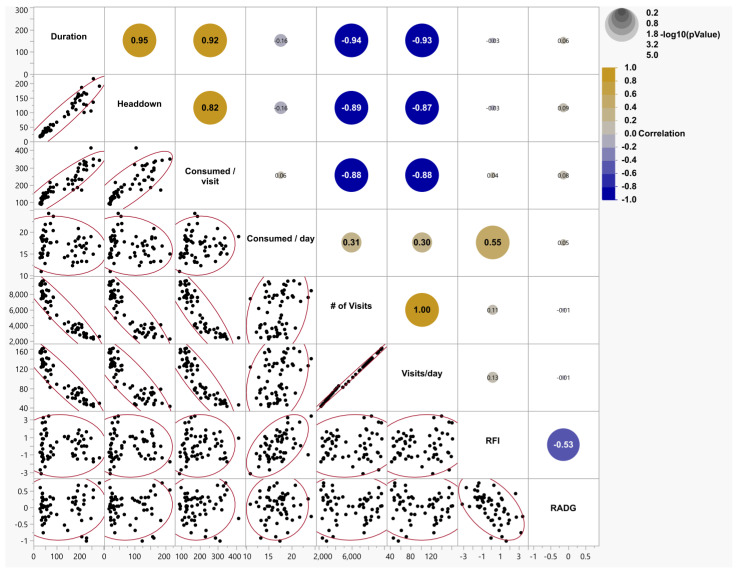
A correlation matrix representing associations among the behavioral and efficiency variables of yearling Angus heifers (*N* = 58). In the upper right portion, the large and bold values are significant simple (unadjusted) correlation coefficients at alpha = 0.05. The color gradient of the circles represents the direction of the correlation, and the size of the circles is proportional [−log 10 (*p*-value)] to the specific *p*-value. The smaller the *p*-value, the larger the circle. The lower left portion has data pairs with 95% bivariate normal density ellipses.

**Table 1 animals-15-03574-t001:** Descriptive details of the animals’ growth performance in the current evaluation. Winter bulls were evaluated from November 2022 to February 2023. Summer bulls were evaluated from June to September of the years 2020, 2021, and 2022. Heifers were evaluated from June to August 2020, and July to September 2022.

	Winter Bulls	Summer Bulls	All Bulls	Heifers
On Test Weight (kg)	369.91 ± 4.47	325.41 ± 4.82	350.35 ± 3.58	278.48 ± 5.01
DOA at end of test (Days)	369.54 ± 1.33	355.94 ± 2.46	363.69 ± 1.35	332.83 ± 3.07
Off Test Weight (kg)	510.66 ± 4.03	446.31 ± 4.85	482.37 ± 3.75	345.64 ± 4.89
Average Daily Gain (ADG) (kg)	1.79 ± 0.01	1.65 ± 0.02	1.73 ± 0.01	1.05 ± 0.04

**Table 2 animals-15-03574-t002:** Descriptive details of the animals’ feeding behavior in the current evaluation. Winter bulls were evaluated from November 2022 to February 2023. Summer bulls were evaluated from June to September of the years 2020, 2021, and 2022. Heifers were evaluated from June to August 2020, and July to September 2022.

	Winter Bulls	Summer Bulls	All Bulls	Heifers
Duration (seconds)	99.69 ± 2.19	107.41 ± 5.14	102.96 ± 2.58	118.09 ± 10.58
Headdown (seconds)	56.37 ± 2.10	62.91 ± 3.71	59.25 ± 2.02	87.74 ± 7.65
Consumed/visit (g/as fed)	220.62 ± 4.30	264.24 ± 9.93	239.80 ± 5.17	195.71 ± 10.61
Consumed/day (kg/as fed)	21.02 ± 0.35	17.47 ± 0.22	20.84 ± 0.28	19.56 ± 0.26
Number of visits (total)	7597.63 ± 121.14	5467.04 ± 169.50	6660.91 ± 122.29	5773.79 ± 329.46
Average visits per day	97.41 ± 1.55	73.56 ±1.93	86.92 ± 1.44	99.98 ± 5.02

## Data Availability

Data available upon request from the authors.

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
