# Peer review of "Associations Between Feeding Behaviors, Residual Feed Intake, and Residual Average Daily Gain in Performance Tested Yearling Bulls and Heifers Fed a High-Forage Diet"

_animals, 2025, doi:10.3390/ani15243574_

Round 1
Reviewer 1 Report
Comments and Suggestions for Authors
The MS contains very good information that catches readers in the factors of feeding behaviour and determinants of animal performance. I suggest authors include a recent citation on residual feed intake in cattle to convince readers. It is also very important to check the English language of the writing by native speakers.
Reviewer 2 Report
Comments and Suggestions for Authors
Dear Editor and Authors,
This study examined the associations between feeding behavior patterns and residual feed intake in yearling Angus bulls and heifers fed high-forage diets, revealing sex-specific correlations between feeding behaviors and feed efficiency metrics. The introduction provides an adequate literature review, and the authors have clearly stated the study's objective. I believe there are several points in the Materials and Methods section that require correction, and that the addition of specific content to the Discussion section would elevate the study to a higher level. Furthermore, it would be beneficial to revise the Conclusion section to ensure consistency with the Discussion and avoid any contradictions. The specific comments I have identified can be found below.
Best regards
Comments:
Although the correlation results in the abstract are presented together with the p-values, we are unable to see the p-values, or their shortened versions indicated with an asterisk, in the correlagrams provided. In order for us to make a better assessment, it is important that the correlations be presented in a way that reflects the p-values, so that we can evaluate whether the correlations are significant.
L131-139: The Analysis section needs to be revised so that it encompasses the parameters included in the correlogram. Although five parameters are presented in this section and in the subsequent ‘Measurements Evaluated’ part, the correlograms contain seven parameters.
I think the addition of a paragraph in the Discussion section that compares the feeding behavior between yearling Angus bulls and yearling Angus heifers would significantly enhance the quality of your study.
Another point concerns the well-established effect of year and season on the live weight of animals. In fact, this phenomenon is clearly evident in Table 1 and Table 2. As can be understood from Table 1, the live weights and ADG values of summer bulls are lower. The authors have provided information regarding the years in which the data were collected in L83-86. At this point, what I am curious about is why you disregarded the effects of season and year, or alternatively, what measures did you take to eliminate or control for these effects?
There appears to be an inconsistency between your discussion and conclusion. In lines 246-247, you mention that 'RADG did not show clear associations with specific feeding behaviors,' yet in the conclusion section , L264-266, you identify RADG as a practical indicator. Could you please clarify this apparent contradiction?
It would be better if the explanations of the abbreviations used under the tables and figures are provided.
Reviewer 3 Report
Comments and Suggestions for Authors
- There's no commentary in the discussion about "ADG". Are 3.8-3.96 kg typical values? Is it a mistake in units?
- There's no characterization and discussion of the analyzed set of behavioral and production traits that form the basis for the calculations.
- Average daily feed intake and RFI data are missing.
- No explanation for "Off Test Weight" and "DOA"
- Why the summer and winter behavior results were put together?
- L244-246 I do not see any evidence for this statement in the results.
Author Response
There’s no commentary in the discussion about ADG. Are 3.8-3.96 kg typical values? Is it a mistake in units?
- These were due to a mistake in units and have been altered to the proper ADG values. These changes can be found in Table 1
There’s no characterization and discussion of the analyzed set of behavioral and production traits that form the basis for the calculations.
- We are unclear as to the question being raised in this comment. Calculations for RFI and RADG are established by Koch et al and the American Angus Association respectively. The behavior traits are averages of time over the testing period determined by the proprietary Vytelle algorithum.
Average daily feed intake and RFI data are missing.
- We have added a new parameter to the study that shows the differences in feed intake consumed daily. This can be found in table 2 and figures 1 and 2 labeled as ‘Consumed/daily’. RFI data is not shown in the table as the average RFI is near zero by design.
No explanation for Off Test Weight and DOA
- DOA means Day of Age and the abbreviation is explained in the list of abbreviations at the end of the paper. Off test weight is the final weight collected at the conclusion of the trial.
Why the summer and winter behavior results were put together?
- These results were put together because we found no statistical differences between summer bulls and winter bulls
L244-246 I do not see any evidence for this statement in the results.
- As shown in Figures 1 and 2, there is an inverse relationship between RFI and RADG. Meaning as RFI decreases RADG increases. The lower RFI the greater the individual animal efficiency and conversely the higher the RADG the greater the efficiency.

Round 2
Reviewer 2 Report
Comments and Suggestions for Authors
Dear Editor and Authors,
The revised manuscript has adequately addressed the issues raised in my earlier review. And the authors' responses are clearly outlined in the attached file. I believe the manuscript is now ready for publication and recommend it be accepted. I also wish to commend the authors for their commitment to this long-term study and for making these findings available to the scientific community. I look forward to their future contributions.
Best regards.